# XPF-ERCC1 protects liver, kidney and blood homeostasis outside the canonical excision repair pathways

Lee Mulderrig[1], Juan I. Garaycoechea[2]*

1 MRC Laboratory of Molecular Biology, Cambridge Biomedical Campus, Francis Crick Avenue, Cambridge, United Kingdom, 2 Hubrecht Institute–KNAW, University Medical Center Utrecht, Uppsalalaan, CT Utrecht, Netherlands

* juan.g@hubrecht.eu

**Data Availability Statement:** All relevant data are within the manuscript and its Supporting Information files

## Abstract

Loss of the XPF-ERCC1 endonuclease causes a dramatic phenotype that results in progeroid features associated with liver, kidney and bone marrow dysfunction. As this nuclease is involved in multiple DNA repair transactions, it is plausible that this severe phenotype results from the simultaneous inactivation of both branches of nucleotide excision repair (GG- and TC-NER) and Fanconi anaemia (FA) inter-strand crosslink (ICL) repair. Here we use genetics in human cells and mice to investigate the interaction between the canonical NER and ICL repair pathways and, subsequently, how their joint inactivation phenotypically overlaps with XPF-ERCC1 deficiency. We find that cells lacking TC-NER are sensitive to crosslinking agents and that there is a genetic interaction between NER and FA in the repair of certain endogenous crosslinking agents. However, joint inactivation of GG-NER, TC-NER and FA crosslink repair cannot account for the hypersensitivity of XPF-deficient cells to classical crosslinking agents nor is it sufficient to explain the extreme phenotype of $Ercc1^{-/-}$ mice. These analyses indicate that XPF-ERCC1 has important functions outside of its central role in NER and FA crosslink repair which are required to prevent endogenous DNA damage. Failure to resolve such damage leads to loss of tissue homeostasis in mice and humans.

## Author summary

The integrity of DNA is essential for life so even the most primitive life forms have evolved a DNA repair 'toolkit' to detect and fix different types of DNA damage. XPF-ERCC1 is an enzyme that can cut DNA and a key player in many of these DNA repair transactions. Consistent with this, inactivating mutations of XPF-ERCC1 in humans and mice lead to a dramatic premature ageing phenotype associated with liver, kidney and bone marrow dysfunction. Here, we ask which of the many functions of XPF-ERCC1 are required to protect tissues from endogenous DNA damage. To do this, we generated cells and mice lacking two of the best characterised functions of XPF-ERCC1: nucleotide excision repair and inter-strand crosslink repair. Surprisingly, neither the cells nor mice lacking these two repair pathways behave like the XPF-ERCC1 mutants, in fact the mice are remarkable

**Funding:** This work was supported by the MRC Laboratory of Molecular Biology. L.M. is supported by a CRUK Program grant awarded to Ketan J. Patel. J.I.G. was supported by the Wellcome Trust Investigator Grant awarded to Ketan J. Patel, King's College, Cambridge and the Hubrecht Institute. The funders had no role in study design, data collection and analysis, decision to publish, or preparation of the manuscript

**Competing interests:** The authors have declared that no competing interests exist.

in their lack of phenotype. Our work suggests that there are functions of XPF-ERCC1 outside of the canonical repair pathways which are important for DNA repair and the homeostasis of multiple organs.

## Introduction

The integrity of DNA is essential for life so even the most primitive life forms have evolved a DNA repair 'toolkit' to detect and remove DNA damage. This damage can range from simple base modifications to breaks in the sugar phosphate backbone, and so specialized pathways exist that are dedicated to fixing specific types of lesions [1].

In mammals, the structure-specific endonuclease XPF-ERCC1 is a key enzyme involved in several of these DNA repair pathways. It is a key component of both branches of nucleotide excision repair (NER), where it removes bulky DNA lesions such as those caused by UV light. Global-genome (GG)-NER senses distortions of the DNA helix whereas transcription-coupled (TC)-NER recognises bulky lesions that lead to stalling of the RNA polymerase. These two branches converge on common NER factors, including XPA, which recruits XPF-ERCC1 to sites of damage. XPF-ERCC1 along with a second endonuclease, XPG, make incisions flanking the DNA adduct prior to filling in of the gap [2].

XPF-ERCC1 is also essential for the repair of inter-strand crosslinks (ICLs) [3,4]. ICLs covalently bind the opposing strands of the DNA helix, blocking both the transcription and replication machineries. ICL-inducing agents like mitomycin C (MMC) and cisplatin are particularly toxic to highly proliferative cells and they are therefore widely utilized in chemotherapy. However, endogenous metabolites like simple aldehydes are also thought to crosslink DNA *in vivo* [5–7]. The majority of ICL repair is thought to occur during replication and it is regulated by a set of proteins defective in the human disease Fanconi anaemia (FA). The FA proteins sense the stalling of the replication fork upon its arrival at the crosslink and monoubiquitinate FANCD2, an event which is absolutely required for incisions at either side of the adduct [8]. This 'unhooking' step is dependent on XPF-ERCC1 and is followed by trans-lesion synthesis (TLS) and homologous recombination (HR) steps to resolve the remaining DNA double-strand break (DSB) [3,4]. Finally, orthologues of XPF-ERCC1 in yeast, flies and plants are involved in the repair of DSBs by HR, and a similar role has been proposed for XPF-ERCC1 in mammals [9].

In humans, defects in GG-NER cause Xeroderma pigmentosum (XP), characterised by photosensitivity and a 10.000-fold increase in the risk of skin cancer [10]. Mutations in TC-NER cause Cockayne syndrome (CS), characterised by photosensitivity, severe growth failure, cachexia, short life span and progressive neurodegeneration, but no cancer predisposition [11]. Children with FA, that results from a defect in ICL repair, suffer from developmental abnormalities, bone marrow failure and have increased risk of cancer [12]. Consistent with the fact that XPF-ERCC1 is involved in multiple DNA repair pathways, inactivating mutations in humans can lead to aspects of XP, CS or FA, and in extreme instances, a combination of all three phenotypes [13–15]. These phenotypes are mirrored in mice, with *Ercc1*$^{-/-}$ (or *Xpf*$^{-/-}$) mice suffering from multisystem degenerative features, severe growth deficits and short life-span [16–18]. The liver of *Ercc1*$^{-/-}$ mice is prominently affected, with hepatocellular karyomegaly which correlates with impaired liver function [16,17]. Rescue of liver failure with a liver-specific *Ercc1* transgene extended the life span of these mice and revealed renal dysfunction coupled with abnormal renal histopathology [19]. Another prominent phenotype in *Ercc1*$^{-/-}$ mice is the development of neurodegeneration, trembling and ataxia, together with kyphosis

and muscle wastage, signs of premature ageing [13]. *Ercc1*[-/-] mice also show a haematopoietic defect in line with their lack of ICL repair [20,21]. In contrast, *Xpa*[-/-] mice are indistinguishable from wild type mice unless treated with carcinogens [22] and FA-deficient mice (like *Fanca*[-/-]) are sterile and develop a mild haematopoietic defect [23,24]

Given the stark contrast between the phenotypes of *Ercc1*[-/-], *Xpa*[-/-] and *Fanca*[-/-] mice, we wanted to determine which aspects of XPF-ERCC1 deficiency result from the joint inactivation of NER and ICL repair, and whether these two pathways act to repair a common lesion *in vivo*. In *E. coli*, NER is essential for ICL repair: the endonuclease $UvrA_2BC$ makes incisions at either side of the crosslink on the same DNA strand, leaving a gap and a large monoadduct on the opposite strand [25,26]. The gap serves as a substrate for HR or can be bypassed by TLS in a *recA*[-] mutant. This pathway is essentially conserved in yeast, where NER factors are absolutely required to unhook the crosslink, allowing subsequent HR or TLS [27,28]. In mammals, both NER branches have been implicated in replication-independent ICL repair, with GG-NER being required for ICL removal during G1 phase of the cell cycle and TC-NER factors being required for transcription-coupled ICL repair [29,30]. Therefore, NER might provide an alternative ICL repair pathway, especially in tissues with low proliferative rates, and cannot therefore rely on replication-coupled repair of crosslinks.

Here we use somatic cell lines and mouse genetics to test the hypothesis that the NER and FA pathways are redundant for ICL repair, shedding light on the physiological relevance of the various functions of the XPF-ERCC1 nuclease. We show that although the TC-NER factors XPA and CSB are involved in cellular protection against ICLs, the cellular sensitivity of XPF-deficient cells to crosslinking agents and the severe phenotype of the *Ercc1*[-/-] mice cannot be explained by joint inactivation of NER and FA repair. Our work suggests that there are functions of XPF-ERCC1 outside of the canonical NER and FA repair pathways which are important for ICL repair and the homeostasis of multiple organs.

## Results

### XPF-ERCC1 is required for normal tissue homeostasis in the liver, kidney and blood

To begin to address the genetic relationship between NER and ICL repair, we first wanted to establish and compare the phenotype of mice lacking NER, the FA pathway or the nuclease XPF-ERCC1. To this end, we used mice lacking FANCA, a component of the FA core complex required for the ubiquitination of FANCD2; XPA, a critical component of both the GG and TC branches of NER; and ERCC1, required for both branches of NER and ICL repair. Disruption of *Ercc1* in an inbred C57BL/6 background yields very few *Ercc1*[-/-] mice, which die soon after birth [31]. However, the severity of the phenotype can be alleviated in an C57BL/6 x 129S6/Sv F1 mixed genetic background and mice die around 12–13 weeks (**S1 Fig**) [17], allowing for a side-by-side comparison of liver, kidney and haemopoietic function across genotypes.

We assessed liver function through biochemical analysis of blood serum and found, in agreement with previous work [16], that *Ercc1*[-/-] mice have raised levels of alanine transaminase (ALT), a marker of liver damage; whereas the *Xpa*[-/-] mice and the *Fanca*[-/-] mice had normal levels (**Fig 1A**). Histologically, only *Ercc1*[-/-] mice displayed karyomegaly in the liver, which is thought to be a consequence of excessive DNA damage (**Fig 1A**). We confirmed that renal function was also affected [19], as shown by increased serum creatinine levels and signs of sclerotic glomeruli in the kidney in *Ercc1*[-/-] mice, but not in *Xpa*[-/-] or *Fanca*[-/-] mice (**Fig 1B**).

Although both *Xpa*[-/-] and *Fanca*[-/-] mice have very mild phenotypes, it is well documented that FA pathway-deficient mice have a reduced number of haematopoietic stem cells (HSCs)

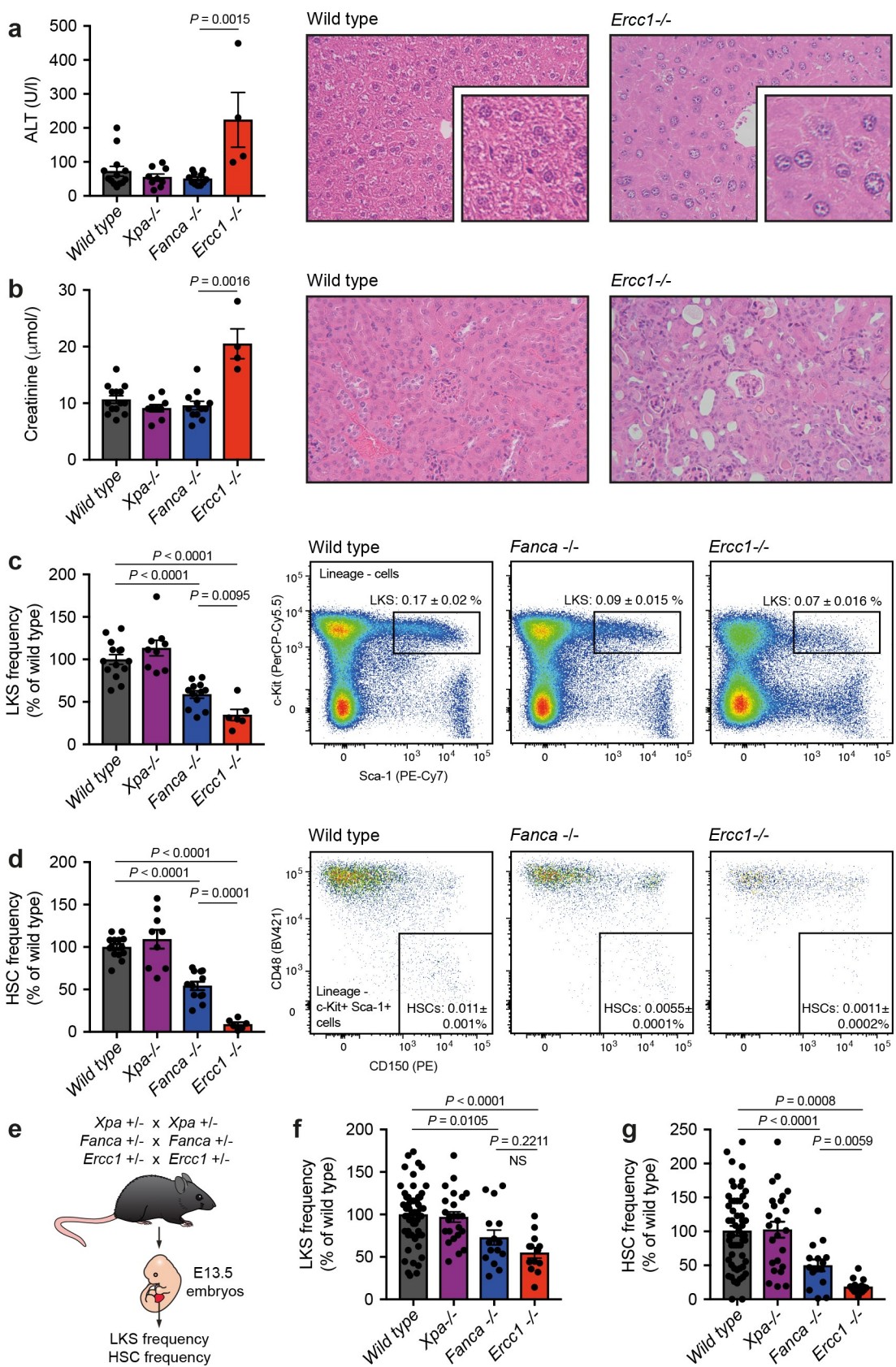

**Fig 1. The nuclease XPF-ERCC1 protects liver, kidney and blood homeostasis. a)** Left: serum levels of alanine transaminase (ALT). Right: haematoxylin and eosin (H&E) staining of liver sections (X200, inset X400). **b)** Left: serum levels of creatinine. Right: H&E staining of kidney sections (X200). **c)** Left: quantification of stem and progenitor cells (lineage- c-Kit+ Sca-1+, LKS) assessed by flow cytometry. Right: representative flow cytometry profiles of 150.000 lineage- cells. **d)** Left: quantification of haematopoietic stem cells (lineage- c-Kit+ Sca-1+ CD48-CD150+, HSC) assessed by flow cytometry. Right: flow cytometry profiles of LKS cells from c). In **a-d**, mice were 8–12 weeks old, C57BL/6 x 129S6/Sv $F_1$ background, n = 13, 9, 11 and 4, error bars represent s.e.m., *P*: two-tailed Mann-Whitney test). **e)** Scheme for the generation of E13.5 embryos for the quantification of HSCs in the foetal liver. **f)** Quantification of LKS cells by flow cytometry in E13.5 foetal liver. **g)** Quantification of HSCs by flow cytometry in E13.5 foetal liver. In **f-g**, pups were E13.5 in a C57BL/6 background, n = 56, 24, 15 and 13, error bars represent s.e.m., *P*: two-tailed Mann-Whitney test.

[32,33]. Whilst young *Xpa*[-/-] mice do not have an overt HSC defect, it has been reported that the haematopoietic progenitor pool is reduced in 1-year old *Xpa*[-/-] mice [20]. *Ercc1*[-/-] mice have been shown to have a severe HSC defect, though this is thought to be due to loss of the FA pathway [20]. We therefore set out to directly compare the haematopoietic defect between *Xpa*[-/-], *Fanca*[-/-] and *Ercc1*[-/-] mice. Firstly, we found that young *Xpa*[-/-] mice had comparable frequencies of progenitor cells (defined as lineage- c-Kit+ Sca-1+ (LKS), **Fig 1C**) and HSCs (LKS CD48- CD150+, **Fig 1D**) when compared to wild type littermates. In contrast, both *Ercc1*[-/-] and *Fanca*[-/-] mice had a significant reduction both in the frequency of LKS cells and HSCs when compared to wild type or *Xpa*[-/-] controls (**Fig 1C and 1D**). Surprisingly, the magnitude of the HSC defect observed in *Ercc1*[-/-] mice was significantly greater than that observed in *Fanca*[-/-] mice (**Fig 1D**, 11.8-fold compared to 1.8-fold).

To exclude the possibility that the more severe HSC defect in *Ercc1*[-/-] mice was being compounded by liver and kidney failure, we next quantified HSCs in embryos. The haematopoietic defect in FA-deficient mice begins *in utero* around day E12.5-E14.5 and should precede liver and kidney dysfunction in *Ercc1*[-/-] mice [34,35]. Therefore, we set up timed matings to generate embryos at E13.5 and quantified the frequency of progenitors and HSC in the foetal liver by flow cytometry (**Fig 1E**). Here we also see that HSC loss is significantly reduced in the *Ercc1*[-/-] mice when compared to *Fanca*[-/-] mice (**Fig 1G and 1F**).

Therefore, lack of ERCC1 not only affects organs that are unperturbed in *Fanca*[-/-] mice and *Xpa*[-/-] mice, but also leads to a more severe contraction in the HSC pool, which is thought to be affected by a deficiency in ICL repair. These data suggest that ERCC1 deficiency removes not only the dominant FA ICL-repair pathway, but also an additional pathway of HSC protection.

## Cells deficient in TC-NER are hypersensitive to crosslinking agents

To shed light on whether the phenotype observed in *Ercc1*[-/-] mice could be due to simultaneous inactivation of both ICL and NER pathways we turned to the haploid somatic cell line HAP1. We used CRISPR/Cas9 to generate a panel of isogenic knock out cell lines for different components of the NER pathway: XPC, that senses helix distortions and initiates GG-NER; CSB, that signals the stalling of the RNA polymerase and triggers TC-NER; XPA, a scaffold protein required by both branches of NER; and XPF, the endonuclease that cleaves DNA on the 5' side of the lesion. The knock out lines were validated by Sanger sequencing (**S2A Fig**), Western blot (**Fig 2A**) and hypersensitivity to UV (**Fig 2B**). We found that haploid *XPF*- cells diploidised spontaneously, so all survival experiments were carried out with diploid lines. We exposed our cell lines to the ICL-inducing agents cisplatin and MMC. In agreement with its documented role in crosslink repair, *XPF*[-/-] cells were extremely sensitive to both agents (**Fig 2C and 2D**). However, *XPA*[-/-] and *CSB*[-/-] cells were also mildly sensitive to these agents, but *XPC*[-/-] cells were indistinguishable from the parental wild type cells. As XPA and CSB both

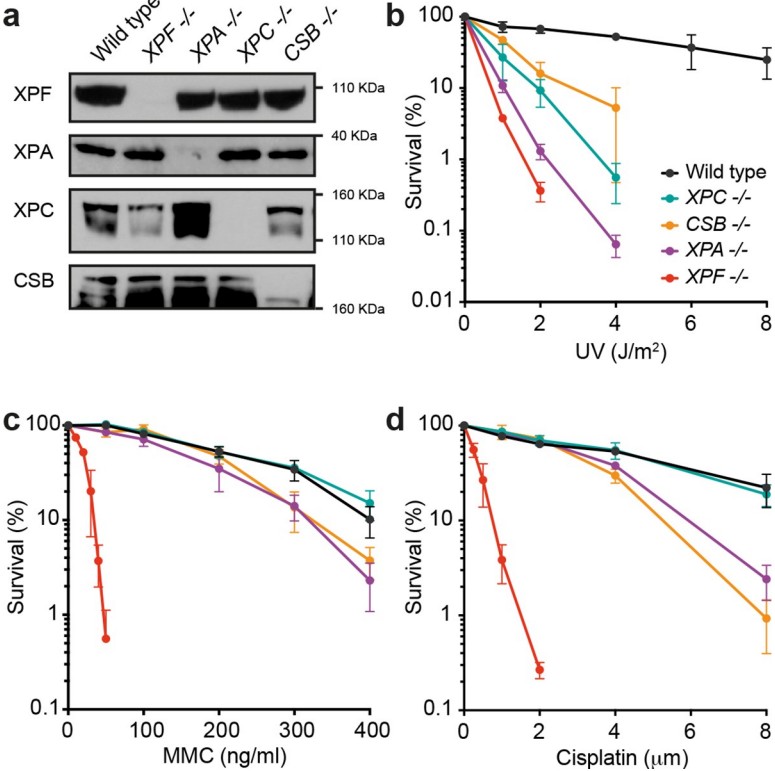

**Fig 2. TC-NER confers protection against ICL-inducing agents. a)** Western blots showing lack of NER proteins in HAP1 knock out lines. See **S2 Fig** for more details. **b)** Survival of HAP1 cells in response to UV light. **c)** Survival of HAP1 cells in response to mitomycin C (MMC). **d)** Survival of HAP1 cells in response to cisplatin. In **b-d**, each data point represents the mean of at least two independent experiments, each carried out in duplicate, error bars represent s.e.m..

operate in the TC branch of NER, this suggests a role for TC-NER, but not GG-NER, in maintaining cellular resistance to crosslinking agents.

## NER and the FA pathway cooperate to repair formaldehyde lesions

Next, we wanted to test whether NER and the FA pathway cooperate to protect against ICLs. To do this, we inactivated FA repair in wild type and $XPA^{-/-}$ cells, using a targeting construct to introduce isogenic disruptions in the *FANCL* locus (**S2B Fig**). FANCL is the E3 ubiquitin ligase responsible for FANCD2 monoubiquitination, which is an essential step in ICL repair [8]. We validated the disruption of the *FANCL* locus by long-range PCR (**S2C Fig**) and detected loss of FANCD2 ubiquitination by western blot (**Fig 3B**). We then quantified the survival of these cells in response to crosslinking agents.

$FANCL^{-/-}$ and $XPF^{-/-}$ lines, both deficient in FA DNA repair, were hypersensitive to the ICL-inducing agents MMC and cisplatin (**Fig 3C and 3D**). However, $XPF^{-/-}$ cells were far more sensitive than FA-deficient cells. We found that $XPA^{-/-}FANCL^{-/-}$ cells were no more sensitive than $FANCL^{-/-}$ cells to MMC, but loss of XPA resulted in additive sensitivity to cisplatin (**Fig 3C and 3D**). This increased sensitivity could be explained by the fact that cisplatin readily generates intrastrand crosslinks (i.e. crosslinking of adjacent bases on the same DNA strand), lesions commonly removed by NER [36]. However, in neither case, joint inactivation of NER and FA repair was sufficient to account for the hypersensitivity of $XPF^{-/-}$ cells. We found that $XPF^{-/-}$ and $XPA^{-/-}FANCL^{-/-}$ cells were equally sensitive to other sources of DNA damage, like

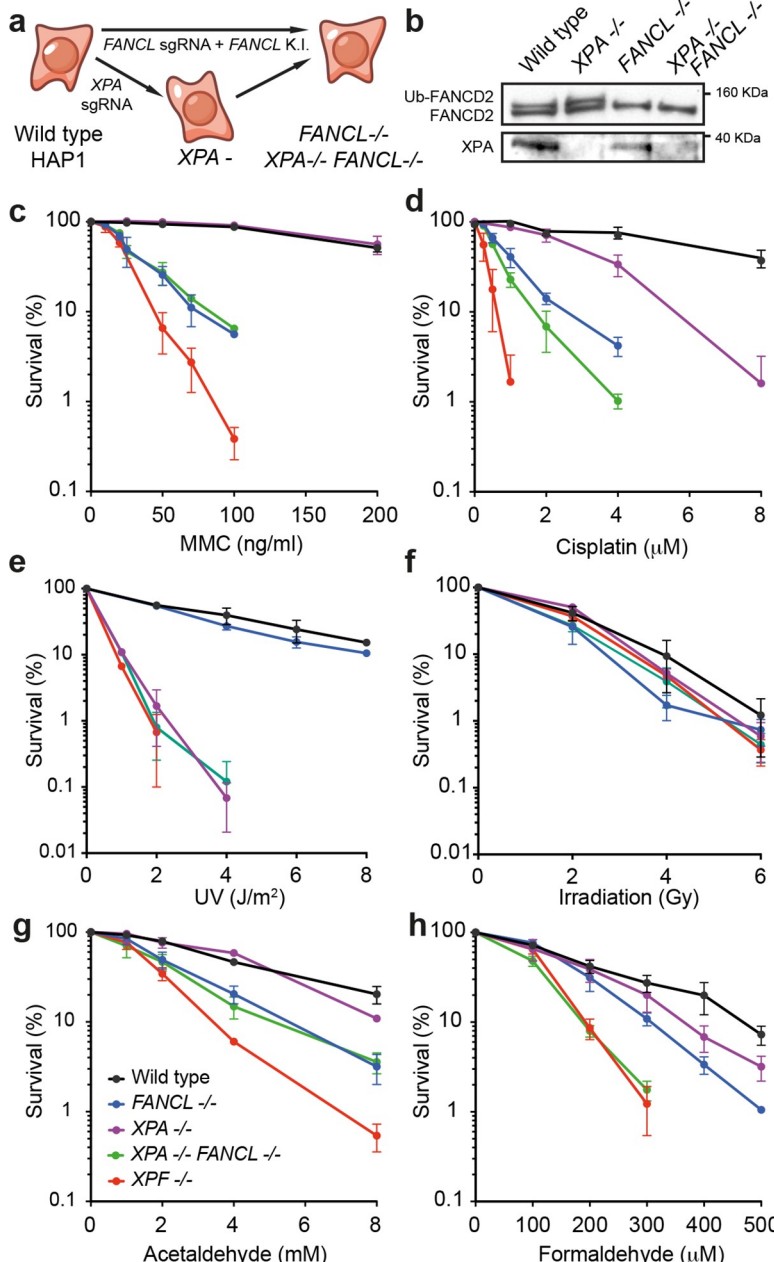

**Fig 3. NER and the FA pathway cooperate to protect cells from formaldehyde, an endogenous crosslinking agent.**
**a)** Scheme for the isogenic disruption of the *FANCL* locus in wild type or *XPA⁻* HAP1 cells. See **S2 Fig** for more details.
**b)** Western blots showing lack of FANCD2 monoubiquitination and XPA protein in HAP1 knock out lines. **c-h)**
Survival of HAP1 cells in response to mitomycin C (MMC), cisplatin, UV, irradiation, acetaldehyde and formaldehyde.
In **c-h**, each data point represents the mean of at least two independent experiments, each carried out in duplicate,
error bars represent s.e.m..

UV light and ionizing radiation, suggesting that the extreme sensitivity of *XPF⁻/⁻* was specific
to ICL-causing agents (**Fig 3E and 3F**). These results highlight that the XPF-ERCC1 nuclease
plays a critical role in ICL repair, perhaps being involved in alternative repair routes, which are
not NER.

Whilst cisplatin or MMC are of significant clinical relevance, we have recently found that reactive aldehydes, by-products of cellular metabolism, could be an important source of endogenous DNA crosslinks [5,7]. These reactive aldehydes may be generated as by-products of metabolism (e.g. 1-carbon metabolism, inflammatory responses), as well as environmental or dietary sources. We therefore exposed $XPA^{-/-}FANCL^{-/-}$ and control cell lines to the simple aldehydes acetaldehyde and formaldehyde (**Fig 3G and 3H**). Lack of NER did not further sensitise $FANCL^{-/-}$ cells to acetaldehyde. In contrast, $XPA^{-/-}FANCL^{-/-}$ cells were more sensitive to formaldehyde than either single mutant, with comparable sensitivity to $XPF^{-/-}$ cells. Although closely related, the reactivity and types of lesions caused by these aldehydes differ [37]. The additional hypersensitivity to formaldehyde may be due to this compound causing a different spectra of DNA lesions, which might include base adducts, DNA inter- and intrastrand crosslinks as well as DNA-protein crosslinks [38]. However, NER and FA repair are both required to maintain cellular resistance to the same endogenous genotoxin—formaldehyde. Although we have exposed cells to exogenous formaldehyde *in vitro*, it is also clear that cells within organisms are exposed to reactive aldehydes without exogenous exposure. Therefore, NER and FA repair constitute alternative pathways to repair lesions caused by this endogenous compound and both rely upon the activity of XPF-ERCC1.

## TC-NER and the FA pathway preserve normal development

We set out to generate mice deficient in both NER and the FA DNA repair pathway to investigate the physiological relevance of this observation. To this end, we first crossed $Xpa^{+/-}$ and $Fanca^{+/-}$ mice in a pure C57BL/6 background. To generate double mutant mice, we exploited the fact that NER-deficient mice are fertile and set up $Xpa^{-/-} Fanca^{+/-}$ x $Xpa^{-/-} Fanca^{+/-}$ crosses, which would generate the highest frequency of double mutants while minimizing the amount of breeding required. We then compared the number of $Fanca^{-/-}$ pups genotyped at 2–3 weeks between the NER proficient and deficient crosses (**Fig 4A and 4B**). We found $Xpa^{-/-}Fanca^{-/-}$ pups to be underrepresented compared to $Fanca^{-/-}$ pups (1.5% instead of 13.3%, $P < 0.0001$) and born as rarely as $Ercc1^{-/-}$ pups in a C57BL/6 background (**S1 Fig**, 1.5% vs 1.9%, $P = 0.67$). This indicated a genetic interaction between NER and ICL repair to preserve mouse development. To investigate the significant reduction in the frequency of double mutants, we performed timed matings between $Fanca^{+/-}$ or $Xpa^{-/-}Fanca^{+/-}$ crosses and sacrificed pregnant females at day E13.5. $Xpa^{-/-}Fanca^{-/-}$ pups were found at a frequency of 11.4% (from n = 35, vs 29% $Fanca^{-/-}$ pups n = 72, $P = 0.04$) and were grossly underdeveloped compared to controls (**Fig 4C**).

Therefore, NER and ICL repair genetically interact to preserve mouse development in a C57BL/6 background. To further dissect this observation, we crossed $Fanca^{+/-}$ mice with mice only lacking GG-NER ($Xpc^{-/-}$) or TC-NER ($Csb^{m/m}$). $Xpc^{-/-}$ mice are sensitive to UV light exposure but have otherwise near-normal lifespans and no overt phenotypes [39]. $Csb^{m/m}$ mice are also sensitive to UV light and show retinal degeneration, as well as mild growth retardation and neurodegenerative changes, an extremely mild version of human CS [40]. We found that $Xpc^{-/-} Fanca^{-/-}$ pups were born at the same frequency as $Fanca^{-/-}$ pups (11.5% vs 13.3%, $P = 0.5616$) (**Fig 4D**). In contrast, $Csb^{m/m} Fanca^{-/-}$ pups were underrepresented compared to $Fanca^{-/-}$ pups (5.5% vs 13.3%, $P = 0.0020$) (**Fig 4E**). When put together, these results show that the FA pathway genetically interacts with components of TC-NER pathway, but not GG-NER, to preserve development. These results agree with the observation that both the FA pathway and TC-NER are required protect cells from crosslinking agents *in vitro* (**Fig 2C and 2D**). In sum, TC-NER and FA repair are jointly required to preserve normal mouse development in a C57BL/6 background.

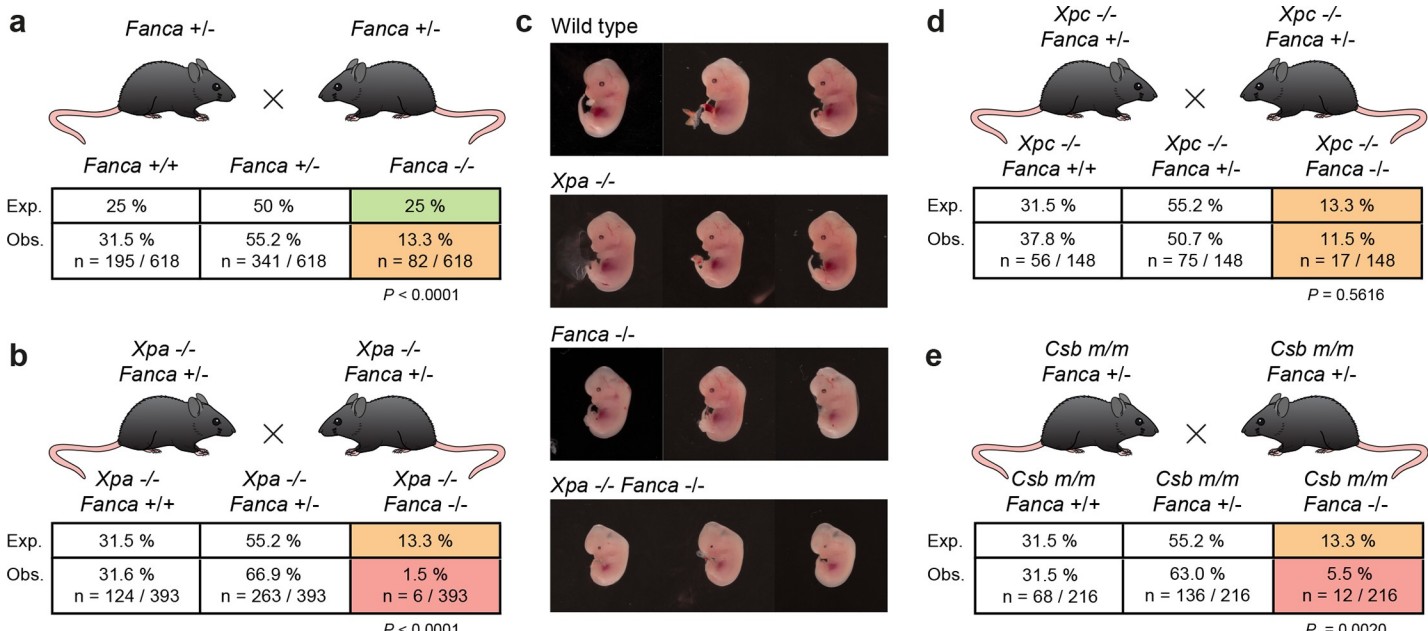

**Fig 4. TC-NER and the FA pathway cooperate to ensure normal mouse development. a)** *Fanca*[+/-] crosses in a C57BL/6 background showing that *Fanca*[-/-] mice are genotyped at sub-Mendelian ratios 2–3 weeks after birth (13.3% instead of the expected 25%, *P*: Fisher's exact test compared to expected numbers. **b)** *Xpa*[-/-]*Fanca*[+/-] crosses for the generation of double mutant mice in a C57BL/6 background **c)** Examination of E13.5 embryos generated from *Fanca*[+/-] or *Xpa*[-/-]*Fanca*[+/-] crosses in a C57BL/6 background. **d)** *Xpc*[-/-] *Fanca*[+/-] crosses for the generation of double mutant mice in a C57BL/6 background. **e)** *Csb*[m/m] *Fanca*[+/-] crosses for the generation of double mutant mice in a C57BL/6 background. In **a, b, d,** and **e,** pups were genotyped 2–3 weeks after birth. In **b, d,** and **e** *P*: Fisher's exact test, compared to numbers from *Fanca*[+/-] crosses.

## Simultaneous inactivation of NER and ICL repair does not recapitulate XPF-ERCC1 deficiency

The strong genetic interaction between TC-NER and FA repair *in utero* prompted us to ask if joint inactivation of these repair pathways might explain some aspects of the *Ercc1*[-/-] phenotype. *Ercc1*[-/-] mice on a C57BL/6 background are born at an extremely low ratio but a C57BL/6 x 129S6/Sv F1 hybrid background circumvents this lethality and allows the study of the role of XPF-ERCC1 in postnatal life (**S1 Fig**) [17]. In complete contrast to the very mild phenotypes of *Csb*[m/m], *Xpa*[-/-] and *Fanca*[-/-] mice, *Ercc1*[-/-] mice suffer from multisystem degenerative features, severe growth deficits and short lifespan. In order to investigate if some aspects of this phenotype are due to joint inactivation of NER and FA repair, we sought to generate *Xpa*[-/-] *Fanca*[-/-] mice on an C57BL/6 x 129S6/Sv F1 hybrid background, and compare this to the phenotype of *Ercc1*[-/-] C57BL/6 x 129S6/Sv F1 mice. Indeed, the embryonic lethality of double mutants in the C57BL/6 background (**Fig 4B**) was completely rescued on the F1 genetic background (**S3A Fig**). The C57BL/6 congenic background is known to potentiate the phenotype of DNA repair-deficient mice [41–43].

Postnatal growth is severely retarded in *Ercc1*[-/-] mice. However, the weight of both *Xpa*[-/-] *Fanca*[-/-] and *Csb*[m/m] *Fanca*[-/-] was undistinguishable from *Fanca*[-/-] mice at 8 weeks of age (**S3B Fig**). We then set out to assess liver and kidney function in these double mutants, as these two tissues lose homeostasis accompanied by morphological changes in *Ercc1*[-/-] mice. The hepatocytes of these mice display polyploidy and this is associated with compromised liver function. We observed no gross histological abnormalities in the liver of *Xpa*[-/-]*Fanca*[-/-] or *Csb*[m/m]*Fanca*[-/-] mutants. Whilst we could detect polyploid nuclei in the liver of *Ercc1*[-/-] mice, we detected normal DNA content in *Xpa*[-/-]*Fanca*[-/-] and *Csb*[m/m]*Fanca*[-/-] livers (**Fig 5A and 5C, S3C Fig**). Liver

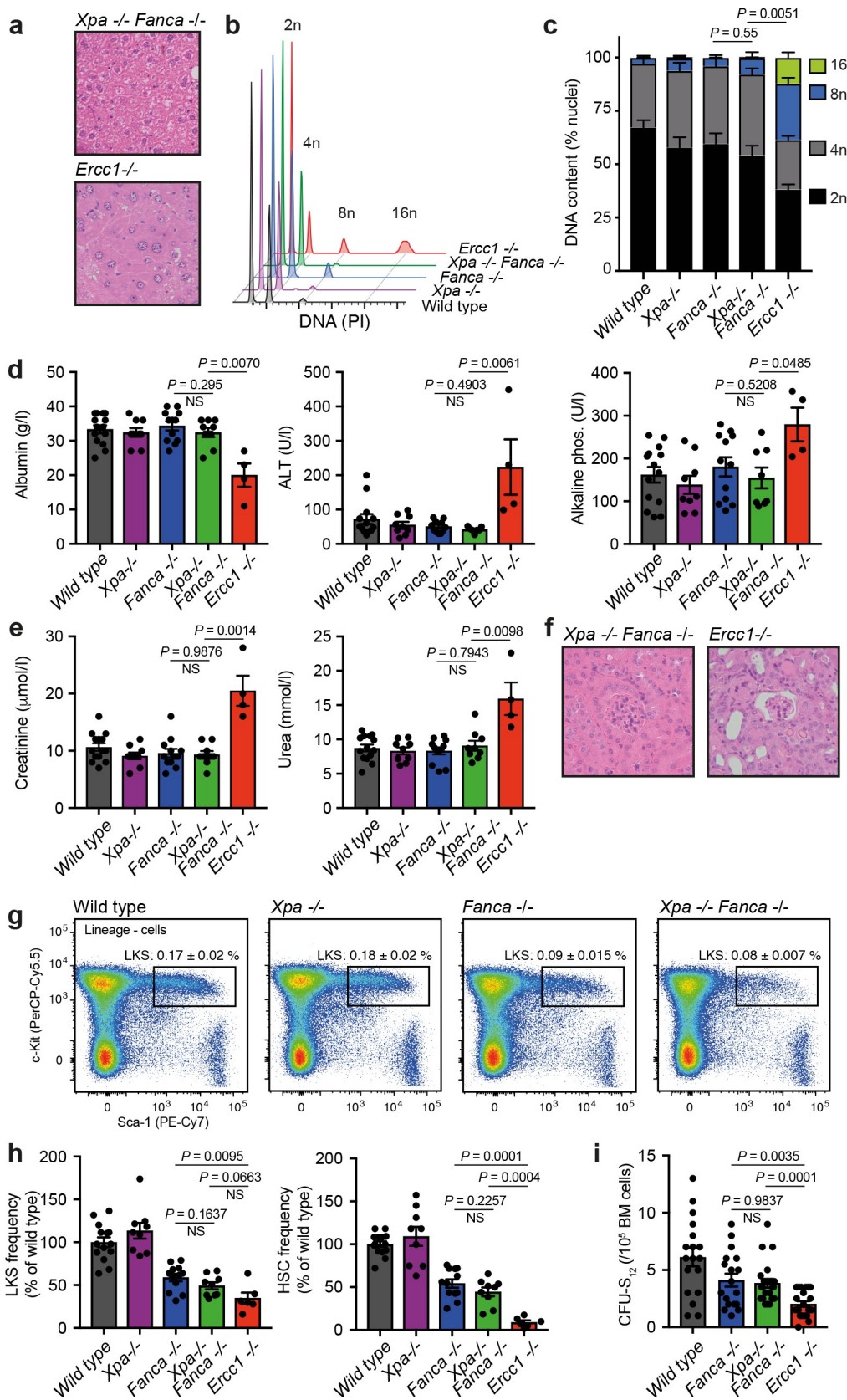

**Fig 5. Joint inactivation of NER and FA crosslink repair does not phenocopy XPF-ERCC1 deficiency. a)** H&E staining of liver sections (X400). **b)** Representative histograms for the flow cytometric analysis of DNA content in the nuclei of liver cells. **c)** Quantification of DNA content in the nuclei of liver cells (*P*: two-tailed Mann-Whitney test for the frequency of 8n nuclei). **d)** Serum levels of albumin, alanine transaminase (ALT) and alkaline phosphatase. **e)** Serum levels of creatinine and urea. **f)** H&E staining of kidney sections (X200). **g)** Representative flow cytometry profiles of 150.000 lineage- cells for the quantification of stem and progenitor cells (lineage- c-Kit+ Sca-1+, LKS). **h)** Quantification of LKS and haematopoietic stem cells (LKS CD48-CD150+, HSC) assessed by flow cytometry. **i)** Quantification of colony forming units–spleen (CFU-$S_{12}$) by transplantation of test bone marrow into irradiated recipients. In **a-i**, mice were 8–12 weeks old, C57BL/6 x 129S6/Sv $F_1$ background, error bars represent s.e.m., *P*: two-tailed Mann-Whitney test. In **c**, n = 6, 5, 5, 5 and 7. In **d, e** and **h**, n = 14, 9, 12, 9 and 4. In **i**, n = 18, 18, 22 and 22.

function was also normal as judged by the concentration of albumin and liver enzymes in blood serum (**Fig 5D**). We then focused our attention on the kidney phenotype, as *Ercc1*$^{-/-}$ mice show abnormal renal histopathology (glomerulosclerosis and protein casts) and renal dysfunction (significantly elevated serum creatinine and urea) [19]. However, we did not detect any of these changes in mice lacking both NER and the FA pathway (**Fig 5E and 5F**, **S3 Fig**).

Finally, we asked if loss of NER could exacerbate the haematopoietic phenotype of *Fanca*$^{-/-}$ mice. We used flow cytometry to quantify progenitor cells and HSCs and observed no difference between *Xpa*$^{-/-}$*Fanca*$^{-/-}$ and *Fanca*$^{-/-}$ controls (**Fig 5G and 5h**). Next, we performed the spleen colony-forming unit assay, which relies on transplantation and is a functional measure of the frequency of stem and progenitor cells; again, we did not detect any difference between the number of *Fanca*$^{-/-}$ and *Xpa*$^{-/-}$*Fanca*$^{-/-}$ progenitors (**Fig 5I**). In all these analyses, *Ercc1*$^{-/-}$ bone marrow was more compromised compared to *Fanca*$^{-/-}$, *Xpa*$^{-/-}$*Fanca*$^{-/-}$ and *Csb*$^{m/m}$*Fanca*$^{-/-}$ mice. Although this more severe haematopoietic defect could be confounded by liver and kidney failure, we showed that the HSC compartment is already contracted *in utero*, prior to severe liver and kidney dysfunction (**Fig 1E–1G**).

In summary, the phenotype of mice with joint inactivation of NER and FA repair in the C57BL/6 x 129S6/Sv F1 background seems indistinguishable from that of *Fanca*$^{-/-}$ controls. This is in complete contrast to the severe phenotype of *Ercc1*$^{-/-}$ mice, which therefore cannot be explained by simultaneous inactivation of NER and ICL repair.

## Discussion

Failure to maintain genome integrity leads to cancer, loss of tissue homeostasis and contributes to ageing. The multifunctional nuclease XPF-ERCC1 is a key component of many DNA repair pathways. Here, we interrogate the functions of XPF-ERCC1 genetically yielding insights into which functions are required to protect cells from ICLs and preserve liver, kidney and bone marrow function. The results presented here show that joint inactivation of NER and the FA pathway do not phenocopy XPF-ERCC1 deficiency, both at the cellular level and in adult mice. These findings shed light on the mechanisms of ICL repair and suggest functions of XPF-ERCC1 outside of the canonical excision pathways (**S4 Fig**).

ICLs are toxic lesions that covalently bind the two strands of DNA together, blocking both transcription and replication. NER removes these adducts in bacteria and yeast. Mammals, however, have evolved the FA pathway, which orchestrates replication-coupled ICL excision. Several cellular and biochemical studies suggest that the NER pathway might also remove ICLs in mammals, outside of replication or during blocked transcription, and that this pathway may act preferentially in certain tissues [29,30]. Here, we first test this hypothesis with an isogenic cellular system, and show that TC-NER provides resistance against classical crosslinking agents (**Fig 2**), in agreement with previous reports using patient-derived lines [44]. However, the FA ICL repair pathway provides the major route of protection against these agents.

Although loss of FANCL completely disables FA repair, we find, surprisingly, that loss of the nuclease XPF-ERCC1 results in much greater hypersensitivity, with $XPF^{-/-}$ cells being far more sensitive to cisplatin and MMC than $FANCL^{-/-}$ cells. Here we show that this increased sensitivity cannot be fully explained by the generation of intrastrand crosslinks or XPF-ERCC1's function in NER, because $XPA^{-/-}FANCL^{-/-}$ cells that lack NER are still not as sensitive as $XPF^{-/-}$ cells (**Fig 3**). These data suggest that the XPF-ERCC1 nuclease has an alternative function in ICL repair. One possibility is that XPF-ERCC1 acts at multiple stages of FA-dependant ICL repair, potentially in the resolution of 'normal' HR intermediates following unhooking of the crosslink, or dealing with 'toxic' DSBs when FA unhooking fails and the replication fork collapses [9,45–47] (**S4 Fig**). The second possibility is that XPF-ERCC1 acts in an entirely separate, FA-independent ICL repair pathway, for example those involving the FAN1 or SNM1A nucleases. In support of this, while SNM1A is non-epistatic with FANCC, SNM1A has an epistatic relationship with XPF-ERCC1, suggesting that these two nucleases function in a common pathway distinct from FA ICL repair [48,49].

We also found that NER and the FA pathway genetically interact to protect cells from formaldehyde, an endogenous source of crosslinks. Interestingly, we did not find a genetic interaction between NER and the FA pathway in response to acetaldehyde, suggesting that these two closely related aldehydes cause a different spectra of DNA lesions. However, this prompted us to investigate if the genetic interaction between NER and the FA pathway extended to mice, to ask if these two pathways cooperate to protect tissues from endogenous DNA damage. Indeed, we find that TC-NER and the FA pathway are jointly required to preserve mouse development in a C57BL/6 isogenic background. It is worth pointing out that $Xpa^{-/-} Fanca^{-/-}$ pups were born at a reduced ratio when compared to $Csb^{m/m} Fanca^{-/-}$, suggesting that while TC-NER is the major NER pathway protecting mouse development in the absence of the FA pathway, there may be additional redundancy between GG-NER and TC-NER (**Fig 4**). However, generation of $Xpa^{-/-} Fanca^{-/-}$ and $Csb^{m/m} Fanca^{-/-}$ on a C57BL/6 x 129S6/Sv F1 background rescued the embryonic lethality and revealed phenotypes indistinguishable from single mutant controls (**Fig 5**). This indicates that NER and FA ICL repair genetically interact, particularly during development, but that this interaction is subtle in adult mice, and heavily affected by genetic background, adding to the list of DNA repair deficient mice whose phenotype is exacerbated in the C57BL/6 background [41–43]. The perinatal lethality of FA-deficient mice, as well as $Ercc1$-/- and $Xpg$-/- mice, depends strongly on genetic background, being worse in C57BL/6. This suggests that modifier loci in the C57BL/6 background potentiate developmental failure in response to DNA damage. Determining the identity of these modifier loci is an interesting avenue for future research and should shed light on cellular and organismal responses to DNA damage.

Nevertheless, the C57BL/6 x 129S6/Sv F1 background allowed us to compare the phenotype of adult mice lacking the XPF-ERCC1 nuclease and those deficient in NER and/or ICL repair. At the organismal level, HSCs are one of the cell populations which are most affected by loss of the FA ICL repair pathway and this observation is true of FA-deficient mice across all genetic backgrounds. Previous studies have suggested a more severe haematopoietic phenotype in ERCC1 hypomorphic mice compared to the haematopoietic phenotype of FA null mice [20,21,50]. Here, for the first time, we provide a side-by-side comparison of $Fanca^{-/-}$ and $Ercc1^{-/-}$ HSC compartments in embryos and adult mice. Importantly, we show that the more severe HSC defect in $Ercc1^{-/-}$ mice is not solely due to joint inactivation of NER and FA ICL repair, because the haematopoietic defect of $Xpa^{-/-}Fanca^{-/-}$ mice is indistinguishable from $Fanca^{-/-}$ mice. This mirrors the greater sensitivity of $XPF^{-/-}$ HAP1 cells compared to $XPA^{-/-} FANCL^{-/-}$ cells in response to crosslinking agents and points to an additional function of XPF-ERCC1 in the protection of HSCs.

Perhaps the most striking observation from our study is the mild phenotype of $Xpa^{-/-}$ $Fanca^{-/-}$ mice, which phenocopying FANCA deficiency, completely contrasts with the severe, complex and short-lived phenotype of $Ercc1^{-/-}$ animals. This observation challenges the prevailing model that joint inactivation of NER and ICL repair pathways greatly contributes to the severe multi-organ failure of XPF-ERCC1 deficiency and, when put together with previous work, draws our attention to the function of this multifunctional nuclease outside the canonical NER and ICL excision pathways. Multiple NER proteins have been attributed additional roles, both in repair pathways outside NER but also in transcription regulation. XPB and XPD are part of TFIIH, an essential component of the general transcription machinery [51]. CSB and XPG, particularly its C-terminus domain, have also been implicated in the regulation of transcription [52–54]. Recent reports have indicated that XPF-ERCC1 also has additional roles in transcription, specifically fine-tuning levels of key target genes through its recruitment of the CCCTC-binding factor (CTCF) chromatin organizer [55–58]. Although, the endonuclease activities of both XPF-ERCC1 and XPG are required to recruit CTCF to chromatin in cell lines [58], XPG-catalytic-dead mice ($Xpg^{E791A}$ and $Xpg^{D811A}$) develop normally and have normal life-spans [59,60]. Therefore, the physiological relevance of XPF-ERCC1-mediated recruitment of CTCF remains unclear.

On the other hand, XPD, XPB, XPG and CSB are also thought to be involved in transcription-coupled repair (TCR). Mutations in these genes cause CS, characterized by hypersensitivity to sun light, cachexic dwarfism, neurodegeneration and features of premature ageing. This complex phenotype cannot be explained by the sole loss of TC-NER, mostly because patients with mutations in $XPA$ do not develop CS [61]. The term transcription-coupled repair (TCR) encompasses well-documented TC-NER as well as poorly-characterised, non-NER (i.e. XPA-independent) functions of proteins mutated in CS. Importantly, inactivation of TCR dramatically exacerbates the mild phenotype of mice lacking GG-NER [62,63]. $Csb^{m/m}Xpa^{-/-}$ and $Csb^{m/m}$ $Xpc^{-/-}$ double mutants show short life span, progressive neurodegeneration and cachectic dwarfism. The fact that loss of CSB can potentiate the phenotype of $Xpa^{-/-}$ mice, already deficient for TC-NER, shows that non-NER TCR has redundant functions to NER. It is not known if the nuclease XPF-ERCC1 also operates in non-NER TCR transactions. However, given the phenotypic overlap between $Csb^{m/m}Xpa^{-/-}$ and $Ercc1^{-/-}$ mice, and the lack of synergistic phenotype of $Xpa^{-/-}Fanca^{-/-}$ mice shown in this study, we propose that a major component of the phenotype of XPF-ERCC1 deficiency results from joint inactivation of NER and non-NER TCR, rather than NER and ICL repair (S4 Fig). Precisely what these TCR transactions entail, as well as the nature of DNA damage, remain poorly defined and should be the focus of future studies.

The clinical heterogeneity of patients with NER deficiency has challenged the DNA repair field for decades. Only by careful genetic dissection can the contribution of the many factors involved begin to be understood. Here, we used clinically relevant mouse models to dissect two of the most comprehensively studied functions of XPF-ERCC1, its roles in NER and ICL repair, and show that deficiency in both these pathways does not result in liver, kidney and more severe haematopoietic defects. Further genetic dissection of the functions of XPF-ERCC1 will be required to address exactly how this key nuclease protects homeostasis in multiple tissues.

## Materials and methods

### Ethics statement

All animal experiments undertaken in this study were with approval of the MRC Laboratory of Molecular Biology animal welfare and ethical review body and the UK Home Office under the Animal (Scientific Procedures) Act 1986 license PFC07716E.

## Mice

All mice were maintained under specific pathogen-free conditions in individually ventilated cages (Techniplast GM500, Techniplast) on Ligno-cel FS14 spruce bedding (IPS, LTD) with environmental enrichment (fun tunnel, chew stick, and Enviro-Dri nesting material (LBS)) at 19–23 ˚C with light from 7:00 am to 7:00 pm and fed Dietex CRM pellets (Special Diet Services) *ad libitum*.

The *Fanca*$^{tm1a(EUCOMM)Wtsi}$ (MGI 4434431, C57BL/6N) and *Ercc1*$^{tm1a(KOMP)Wtsi}$ (MGI 4362172, C57BL/6) alleles have been described previously [31,33]. *Xpa*$^{tm1Hvs}$ (MGI 1857939, C57BL/6), *Ercc6*$^{tm1Gvh}$ *(Csb*$^m$, MGI 1932102, C57BL/6*)* and *Xpc*$^{tm1Ecf}$ (MGI 1859840, C57BL/6) mice were described previously and a kind gift from G.T. van der Horst, Errol Friedberg and Jan Hoeijmakers [22,39,40]. To generate *Xpa*$^{-/-}$*Fanca*$^{-/-}$, *Xpc*$^{-/-}$*Fanca*$^{-/-}$ and *Csb*$^{m/m}$*Fanca*$^{-/-}$ mice on a pure C57BL/6 background, *Fanca*$^{+/-}$ mice were crossed with *Xpa*$^{+/-}$, *Xpc*$^{+/-}$ and *Csb*$^{+/m}$, respectively. From the resulting progeny we intercrossed NER$^{+/-}$*Fanca*$^{+/-}$ to generate all possible genotypes. To further bias the breeding for the generation of double mutants, we intercrossed mice that were both NER$^{-/-}$*Fanca*$^{+/-}$.

To generate mice in a C57BL/6 x 129S6/Sv F$_1$ background, the various C57BL/6 alleles were first backcrossed onto the 129S6/Sv background: *Fanca*$^{+/-}$ and *Csb*$^{+/m}$ 10 generations, *Xpa*$^{+/-}$ 6 generations and *Ercc1*$^{+/-}$ 7 generations. For *Fanca*$^{-/-}$ and *Ercc1*$^{-/-}$ F$_1$ mice, C57BL/6 heterozygous mice were crossed with 129S6/Sv heterozygous mice. For *Xpa*$^{-/-}$*Fanca*$^{-/-}$ and *Csb*$^{m/m}$*Fanca*$^{-/-}$ F$_1$ mice, *Fanca*$^{+/-}$ 129S6/Sv were crossed with *Xpa*$^{+/-}$ or *Csb*$^{+/m}$ to generate double heterozygous mice, these mice were then intercrossed to generate *Xpa*$^{-/-}$*Fanca*$^{+/-}$ and *Csb*$^{m/m}$*Fanca*$^{+/-}$ 129S6/Sv which were crossed with *Xpa*$^{-/-}$*Fanca*$^{+/-}$ and *Csb*$^{m/m}$*Fanca*$^{+/-}$ C57BL/6, respectively. For the phenotyping of C57BL/6 x 129S6/Sv F$_1$ mice, mice were used between 8–12 weeks.

Embryos were generated in a C57BL/6 background and analysed at day E13.5 of development. Females used in timed matings were used between 8 and 18 weeks old.

## Cell lines

HAP1 cells (Haplogen) were purchased from Horizon Discovery and cultured in IMDM medium (Gibco) supplemented with 10% dialysed foetal calf serum and penicillin/streptomycin. Cells were grown at 37 ˚C and 5% $CO_2$. All cell lines used in the study were tested to be mycoplasma-free.

## CRISPR/Cas9-mediated gene disruptions in HAP1 cells

Guide sequences for each gene disruption can be found in **S1 Table**. Plasmids containing each pair of guide sequences were obtained from the Wellcome Trust Sanger Institute. HAP1 cells were transfected with the vector containing guides along with the Cas9 containing PX461 vector using Turbofectin (Origene). Two days post-transfection, GFP+ cells were single-cell sorted into 96-well plates containing medium supplemented with 20% foetal calf serum, using a MoFlo cell sorter (Beckman-Coulter). After 14 days of incubation at 37 ˚C, individual clones were analysed for expression of the relevant protein by western blotting. Targeted loci were subjected to Sanger sequencing (GATC). **S2 Table** contains the primers used to amplify the relevant loci by PCR and **S3 Table** contains the primers used for Sanger Sequencing. For FANCL targeting, LR-PCR was used to screen clones with correct integration of the targeting construct, the primers can be bound in **S2 Table**.

## Western blotting

Cells were lysed for 30 min on ice in RIPA buffer (Thermo Fisher Scientific), including protease inhibitor cocktail (Roche) and phosphatase inhibitor cocktail (Roche). For the detection of

FANCD2 in HAP1 cells, cells were treated with MMC 500 ng/ml overnight and protein samples were run on a 3–8% Tris-Acetate gel (Thermo Fisher Scientific). Samples were blotted to a 0.45 μm nitrocellulose membrane. Protein samples were run on a 4–12% Bis-Tris gel (Thermo Fisher Scientific) to detect XPC, XPA, XPF and CSB in HAP1 cells. Antibodies used were anti-XPC (D1M5Y, Cell Signaling, 1:1000), XPF (D3G8C, Cell Signalling, 1:1000), XPA (D9U5U, Cell Signalling, 1:1000), CSB (ab96089, abcam, 1:1000) and FANCD2 polyclonal antisera (1:3000) [64].

## Colony survival assay

Haploid HAP1 cells were allowed to diploidise spontaneously and enriched for diploid cells based on DNA content by flow cytometry. All colony survival assays were carried out with diploid HAP1 cells, which were trypsinised and resuspended at a concentration of $2x10^5$ cells/ml. Drugs (mitomycin C, cisplatin, acetaldehyde or formaldehyde) were added at a 2X concentration in a total volume of 2 ml and incubated for 2 hours at 37˚C. After 2 hours, two 1/10 serial dilutions were made and 100 μl cells were plated onto a 6 well plate-containing 5 ml IMDM supplemented with 10% dialysed foetal calf serum and penicillin/streptomycin. For UV and X-ray irradiation, cells were diluted to $1x10^5$ cells/ml in 1 ml PBS, irradiated in a 6 well plate, and then immediately after irradiation, cells were spun down and resuspended in 1 ml IMDM and two 1/10 serial dilution were made and 100 μl cells were plated onto 6 well plates. Cells were grown at 37˚C for 7 days. For visualization, colonies were washed with PBS and then stained with 6% v/v gluteraldehyde containing 0.5% crystal violet for 1 hour before washing again with PBS.

## Histology

Histological analysis was performed on tissues that had been fixed in neutral buffered formalin for 24h. The samples were paraffin embedded and 4 μm sections were cut before staining with haematoxylin and eosin.

## Nuclei isolation and DNA content analysis

Liver and kidneys were dissected and passed through a 40-μm filter. Cells were washed twice in LA buffer (250 mM sucrose, 5 mM $MgCl_2$ and 10 mM Tris-HCl, pH 7.4). After washing, the cell pellet was resuspended in 1 ml of buffer LB (2 M sucrose, 1 mM $MgCl_2$ and 10 mM Tris-HCl, pH 7.4) and centrifuged at 16000g for 30 minutes. The white nuclei-containing pellet was resuspended in LA buffer and kept on ice for analysis. For DNA content analysis, nuclei were fixed drop-wise in cold 96% ethanol. Nuclei were pelleted and re-suspended in 400 μl of PBS. Propidium iodide solution (Sigma) was added at a final concentration of 40 μg/ml together with Ribonuclease A (Sigma) at a final concentration of 100 μg/ml. The samples were incubated on ice for one hour and then analysed on LSRII flow cytometer (BD Pharmingen). The data was analysed with FlowJo 10.0.6 (Tree Star).

## Serum biochemistry

Serum was collected from 200 μl of whole blood into Microvette 200 conical tubes (MCV200-SER) after centrifugation. Levels of urea, creatinine, aspartate aminotransferase, albumin, and alkaline phosphatase of serum samples were measured using a Siemens Dimension RxL analyser.

## HSPC analysis by flow cytometry

Bone marrow cells were isolated from the femora of mutant mice and aged-matched controls by flushing cells and passing them through a 70-μm filter. The following antibodies were used to stain for HSCs: FITC- conjugated lineage cocktail with antibodies anti-CD3e (clone 145-2C11, eBioscience), CD4 (clone H129.19, BD Pharmingen), CD8a (clone 53–6.7, BD Pharmingen), CD11b/Mac-1 (clone M1/70, BD Pharmingen), CD11c (clone N418, eBioscience), Ly-6G/Gr-1 (clone RB6- 8C5, eBioscience), B220 (clone RA3-6B2, BD Pharmingen), FcεR1α (clone MAR-1, eBioscience), TER-119 (clone Ter119, BD Pharmingen), CD41 (clone MWReg30, BD Pharmingen); anti-c-Kit (PerCP-Cy5.5, clone 2B8, eBioscience), anti-Sca-1 (PE-Cy7, clone D7, eBioscience), anti-CD48 (biotin, clone HM48-1, BioLegend) and anti-CD150 (PE, clone TC15- 12F12.2, BioLegend). After staining for 15 minutes in PBS + 2% FCS, the cells were washed and incubated with streptavidin conjugated to Brilliant Violet 421 (Bio-Legend) for another 15 minutes.

Foetal livers from E13.5 embryos were dissected and placed in 1 ml of PBS + 2% FCS. The foetal livers were triturated gently using a P1000 pipette until a homogenous suspension was formed. The cells were then passed through a 40 μm cell strainer (Falcon) and nucleated cells were counted with 3% acetic acid on a Vi-Cell XR cell viability counter (Beckman Coulter). $10 \times 10^6$ foetal liver cells were spun down for 5 minutes at 1200 rpm and stained as above.

## Colony-forming unit spleen (CFU-S$_{12}$) assay

Mice were sacrificed between 8 and 12 weeks and 1 or $2 \times 10^5$ nucleated bone marrow cells were injected intravenously into recipient mice that had been irradiated with 8 Gy split between two equal 4 Gy doses 4 hours apart. 12 days after transplantation the mice were sacrificed, the spleens were fixed in Bouin's solution (Sigma) for at least 24 hours and the gross colonies were enumerated and expressed relative to the number of nucleated bone marrow cells injected.

## Supporting information

**S1 Fig. Generation of *Ercc1*-/- mice. a)** *Ercc1*$^{+/-}$ crosses in a C57BL/6 background showing that *Ercc1*$^{-/-}$ mice are genotyped at sub-Mendelian ratios 2–3 weeks after birth (1.9% instead of the expected 25%, Fisher's exact test: $P < 0.0001$). **b)** *Ercc1*$^{+/-}$ crosses to generate *Ercc1*$^{-/-}$ mice in a C57BL/6 x 129S6/Sv F$_1$ background. Although *Ercc1*$^{-/-}$ mice are genotyped at sub-Mendelian ratios 2–3 weeks after birth (12.5% instead of the expected 25%, Fisher's exact test: $P = 0.0002$), the mixed genetic background rescues the lethality observed in the C57BL/6 background. **c)** Survival of *Ercc1*$^{-/-}$ and control mice in C57BL/6 and C57BL/6 x 129S6/Sv F$_1$ genetic backgrounds.
(TIF)

**S2 Fig. Generation of HAP1 knock out lines. a)** Maps of human NER genes. The inset shows the nucleotide and predicted amino acid sequence of wild type and knock out HAP1 lines, generated by introducing deletions with CRISPR/Cas9, see Methods for details. **b)** Map of the human *FANCL* gene and targeting of exon 8 to generate isogenic disruptions in *FANCL*. **c)** Agarose gel showing disruption of the *FANCL* locus by long-range PCR. Primers hybridise outside the homology arms and within the targeting construct.
(TIF)

**S3 Fig. Joint inactivation of TCR and FA crosslink repair does not phenocopy XPF-ERCC1 deficiency. a)** Crosses for the generation of double mutant mice in a C57BL/6 x 129S6/Sv F$_1$ background, pups were genotyped 2–3 weeks after birth (*P*: Fisher's exact test, compared to expected numbers). **b)** Weights of 8-week-old females in a C57BL/6 x 129S6/Sv F$_1$ background

(*P*: two-tailed Mann-Whitney test, n = 9, 3, 5, 3, 4, 5 and 7). **c)** Quantification of DNA content in the nuclei of liver cells (*P*: two-tailed Mann-Whitney test for the frequency of 8n nuclei, n = 6, 5, 5, 5 and 7). **d)** Serum levels of albumin, alanine transaminase (ALT) and alkaline phosphatase. **e)** Serum levels of creatinine and urea. **f)** Quantification of stem and progenitor cells (lineage- c-Kit+ Sca-1+, LKS) and haematopoietic stem cells (LKS CD48- CD150+, HSC) assessed by flow cytometry. In **d-f**, mice were 8–12 weeks old, C57BL/6 x 129S6/Sv $F_1$ background, error bars represent s.e.m., *P*: two-tailed Mann-Whitney test. n = 14, 6, 12, 6 and 4. (TIF)

**S4 Fig. Model of the interactions between DNA repair pathways that converge on the nuclease XPF-ERCC1. a)** NER and FA ICL repair interact to protect against certain crosslinkers *in vitro* and to ensure normal development in a C57BL/6 background. **b)** XPF-ERCC1 has a role in ICL repair outside NER and ICL unhooking, potentially the repair of DNA double strand breaks (DSBs). **c)** The XPF-ERCC1 phenotype is likely due to deficiency in NER and non-NER TCR rather than NER and FA ICL repair. How exactly XPF-ERCC1 operates in TCR, and the nature of this damage, remains to be established. (TIF)

**S1 Table. gRNAs for the generation of CRISPR knock outs.** (DOCX)

**S2 Table. Oligos for the screening of CRISPR knock outs by PCR.** (DOCX)

**S3 Table. Oligos for Sanger sequencing of PCR products.** (DOCX)

# Acknowledgments

We are grateful to Ketan J. Patel for creating a stimulating research environment, and giving us the freedom, support and resources to pursue our own ideas. We thank Errol Friedberg, G. T. van der Horst and Jan Hoeijmakers for sharing the *Xpc*$^{-/-}$, *Xpa*$^{-/-}$ and *Csb*$^{m/m}$ mice. We thank the ARES staff, Biomed and the LMB Genotyping service for their help with mouse work; and members of the Patel and Crossan laboratories for critical reading of the manuscript. The Human Research Tissue Bank (supported by the NIHR Cambridge Biomedical Research Centre) processed histology.

# Author Contributions

**Conceptualization:** Juan I. Garaycoechea.

**Data curation:** Juan I. Garaycoechea.

**Investigation:** Lee Mulderrig, Juan I. Garaycoechea.

**Writing – original draft:** Juan I. Garaycoechea.

**Writing – review & editing:** Lee Mulderrig, Juan I. Garaycoechea.

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
