## [Decision Letter · Decision Letter 0]

3 Nov 2019

Dear Dr Garaycoechea,

Thank you very much for submitting your Research Article entitled 'XPF-ERCC1 protects liver, kidney and blood homeostasis outside the canonical excision repair pathways' to PLOS Genetics. Your manuscript was fully evaluated at the editorial level and by independent peer reviewers. The reviewers appreciated the attention to an important topic but identified some aspects of the manuscript that should be improved.  Revision will require a more thorough discussion that addresses some very thoughtful questions raised by the reviewers, but you need not include new experimental results unless you opt to do so.

We therefore ask you to modify the manuscript according to the review recommendations before we can consider your manuscript for acceptance. Your revisions should address the specific points made by each reviewer.

[LINK]

Yours sincerely,

Nancy Maizels, Ph.D.

Associate Editor

PLOS Genetics

Gregory Barsh

Editor-in-Chief

PLOS Genetics

Reviewer's Responses to Questions

**Comments to the Authors:**

Reviewer #1: Mulderrig and colleagues present a descriptive genetic study of the interaction between two key pathways thought to defend against DNA interstrand crosslinks (ICLs) in mammals, namely nucleotide excision repair (NER) and factors deficient in Fanconi anemia (FA). While little in the way of novel mechanism is revealed, much can be surmised from the detailed phenotypic analysis, and it is my view that this is an important and timely contribution. The reliance on the transcription-coupled aspects of NER (TC-NER) have not been previously addressed or demonstrated in a systematic fashion. The MS is well-written, the data support the conclusions, the experiments are relatively thorough and the work is generally of good quality.

My only comments are as follows:

1. It has been suggested on a number of occasions that some rodent cell types are deficient, or at least show reduced, GG-NER, possibly due to low DDB2 levels, originally referred to as the 'rodent repairadox'. It remains unclear for many cell types whether this is really the case in vivo, but the possibility that certain rodent cells types could be less efficient in performing GG-NER than perhaps human cells are should be acknowledged and its ramifications briefly discussed.

2. In the Discussion section, few alternative suggestions are proposed for the identity of the XPF-ERCC1 dependent ICL repair pathway(s) that act in addition to TC-NER and the FA pathways. It would be nice to see the authors 'stick their necks out' and speculate a little on this point: could it/they be FAN1, SNM1A or mismatch repair-related?

Reviewer #2: Loss of the XPF/ERCC1 nuclease complex results in a more profound phenotype than absence of other nucleotide excision repair (NER) genes. Ercc1-/- mice are born at substantially less than expected frequencies and die shortly after birth. In humans, complete inactivation of either XPF or ERCC1 is incompatible with viability. Although the basis of the apparently absolute requirement for XPF/ERCC1 remains uncertain, it is known that the complex is important for multiple DNA transactions: NER (both global and transcription coupled), but also interstrand crosslink (ICL) repair via the Fanconi Anemia (FA) pathway. Additionally, ERCC1/XPF has a function in telomere maintenance, contributes to repair of double strand breaks in both homology directed and end joining pathways, and is involved in the repair of abasic lesions. Presumably the profound phenotype of XPF/ERCC1 deficiency reflects diminished efficacy of the collective of these, or as yet unidentified, pathways.

In an effort to clarify the basis for the extreme phenotype the authors of this submission have examined the response to various DNA damaging agents, including acetaldehyde, formaldehyde, cisplatin and MMC in cells either lacking ERCC1, or XPA (representing loss of both GG and TC-NER), or FANCL (a key FA gene), or both XPA and FANCL. XPA and the FANCL double knockout cells were not as sensitive to cisplatin, MMC, or acetaldehyde as ERCC1-/- cells. These results emphasized the greater pathology of Ercc1-/- as compared to loss of both NER pathways, or the FA pathway, or the combined pathways. They argued that Ercc1 performed repair functions additional to those of NER and FA. These results and conclusion contrasted with the results with formaldehyde treated cells in which the FANCL-/-XPA-/- cells were just as sensitive as the ERCC1-/- cells.

The authors also studied the effect of combined NER and FA deficiency on mouse development. Xpa-/- Fanca-/- C57BL/6 pups were born as infrequently as Ercc1-/- animals. This observation was further dissected by examining pup frequencies in Fanca-/- Xpc-/- or Fanca-/- Csb-/- C57BL/6. Csb (mutant in Cockayne Syndrome and required for TCR) was chosen because it restricted the deficiency to TCR. The absence of Xpc in the Fanca-/- background did not affect birth frequencies, while Csb deficiency reduced them by half, significant, but not as profound as the Ercc1-/- or Xpa-/- Fanca-/- reduction. This result focused attention on the requirement for TC-NER and FA in normal development, but implied that additional ERCC1 pathways were also important.

The analyses of Ercc1 knockout was also performed in F1 hybrids of C57BL/6 crossed with 129S6/Sv, a strain that suppresses the developmental severity of Ercc1 loss. The role of the NER and FA pathways in postnatal life was examined in the F1 hybrids. Xpa-/- Fanca-/- pups were born at normal frequencies and examination of liver, kidney, and hematopoietic tissues indicated that the Ercc1-/- animals were more severely compromised than either the Xpa-/-Fanca-/- or Csb-/-Fanca-/- animals.

In their Discussion the authors cite work from van der Pluijm et al, who studied Csb and Csb/Xpa deficient C57BL/6 mice, whose birth frequencies were reduced but not as severely as Ercc1-/- mice.

Critique:

This paper presents a series of thoughtfully designed and detailed analyses to address the question: do the NER and FA pathways account for the activities of XPF/ERCC1? The answer seems to be: it depends on the specifics of the assay. This rather mixed conclusion, in a narrative that twists and turns, should be clarified with some additional data and discussion. Otherwise it seems to be two parallel reports-one describing results with one damaging agent and mouse strain that conclude the affirmative, while the other presents the contrary view based on assays with different damaging agents and mouse background.

1. In the experiments with HAP1 knockout cells the authors found that the XPA/ FANCL double KO cells were as sensitive to formaldehyde as the XPF KO. Consequently, the elements of the XPF pathway that conferred protection to formaldehyde were duplicated by NER and FA. In contrast, the sensitivity of XPF KO cells to MMC, cisplatin, and acetaldehyde was not completely captured by XPA/FANCL KO. This implied that while these compounds generated adducts repaired by NER and FA, they also produce toxic structure (s), not formed by formaldehyde, that is/are resistant to NER and FA, but resolved by XPF. XPF/ERCC1 contributes to various forms of double strand break (DSB) repair. Recent work in yeast (Cell Cycle 16, 45, 2017) suggest an involvement in homologous recombination pathways in the repair of acetaldehyde damage. Sensitivity to crosslinking agents is a mark of many HR deficiencies. The authors raise the possibility of ERCC1 participation in the repair of DSBs in the context of the compound sensitivity experiments. Do cells treated with these compounds show evidence for DSBs, perhaps replication related, that formaldehyde treated cells do not? Some simple immunofluorescence analyses or comet assays would address this.

2. The adverse effects of FA deficiencies are interpreted as reflecting the activity of the pathway in ICL repair. Although this is reasonable, there is a school of thought that adds sensitivity to inflammatory mediators as an important aspect of FA. This is a feature that might have profound consequences during development and should be mentioned.

3. In C57BL/6 mice Xpa/Fanca KO recapitulates the low birth yields seen with Ercc1 KO mice. Consequently, it appears that the NER and (at least some aspect of) the FA pathways substitute for Xpf/Ercc1 during early mouse development. The authors discuss this and conclude that the combination of the TCR and FA pathways “preserve development”. However, the decline in live births in the Csb/Fanca pups is not as great as in the Xpa/Fanca background. Is there another, non NER, role for XPA?

4. The results of with C57BL/6 were confounded by the suppression of effect of the Xpa/Fanca knock out in the C57BL/6 x 129S6/Sv cross. The experiments with the crossed animals are more difficult to interpret, given the amelioration of phenotypes by unknown modifiers supplied by 129S6/Sv. The operating assumption with the F1 hybrid is that the “modifying” component does not affect the deficiency under study. For example, the results indicate that Ercc1 loss is more consequential than Xpa/Fanca for various tissue markers, in the F1. Is it possible that the 129S6/Sv strain supplies a suppressor of FA deficiency, making the Xpa/Fanca KO F1 strain still functionally Xpa-/- but with diminished influence of FA inactivity? Are the cells from the F1 “FA-/-“ as defective in Fanconi pathway functions (like ICL repair) as C57BL/6 “FA-/- “? Pace et al demonstrated in a well known study (Science 2010) that Ku70 ”corrupts DNA repair in the absence of the FA pathway”. Loss of Ku reduced sensitivity to ICLs and chromosome instability. Does the 129S6/Sv strain contribute something like this to the F1?

5. In the end the authors reach the conclusion that the lack of TCR, of both NER and non NER lesions, accounts for much of the phenotype of XPF-ERCC1 deficiency. While that may make a central contribution, the results across the study are so influenced by experimental design so as to restrict conclusions to the specific conditions of the analysis.

**Have all data underlying the figures and results presented in the manuscript been provided?**

Reviewer #1: Yes

Reviewer #2: Yes

PLOS authors have the option to publish the peer review history of their article (what does this mean?). If published, this will include your full peer review and any attached files.

Reviewer #1: No

Reviewer #2: No

---

## [Editor Report · Decision Letter 1]

5 Dec 2019

Dear Juan,

We are pleased to inform you that your manuscript entitled "XPF-ERCC1 protects liver, kidney and blood homeostasis outside the canonical excision repair pathways" has been editorially accepted for publication in PLOS Genetics. Congratulations!

Yours sincerely,

Nancy Maizels, Ph.D.

Associate Editor

PLOS Genetics

Gregory Barsh

Editor-in-Chief

PLOS Genetics

Comments from the reviewers (if applicable):

**Data Deposition**

http://datadryad.org/submit?journalID=pgenetics&manu=PGENETICS-D-19-01655R1

**Press Queries**

---

## [Editor Report · Acceptance letter]

17 Dec 2019

PGENETICS-D-19-01655R1 

XPF-ERCC1 protects liver, kidney and blood homeostasis outside the canonical excision repair pathways 

Dear Dr Garaycoechea, 

We are pleased to inform you that your manuscript entitled "XPF-ERCC1 protects liver, kidney and blood homeostasis outside the canonical excision repair pathways" has been formally accepted for publication in PLOS Genetics! Your manuscript is now with our production department and you will be notified of the publication date in due course.

With kind regards,

Nicholas White

PLOS Genetics

On behalf of:
